# Experimentally Validated Extension of the Operating Range of an Electrically Driven Turbocharger for Fuel Cell Applications

**Markus Schoedel** *,†, **Marco Menze** † and **Joerg R. Seume**

Institute of Turbomachinery and Fluid Dynamics, Leibniz Universitaet Hannover, An der Universitaet 1, 30823 Garbsen, Germany; menze@tfd.uni-hannover.de (M.M.); seume@tfd.uni-hannover.de (J.R.S.)
* Correspondence: schoedel@tfd.uni-hannover.de
† These authors contributed equally to this work.

**Abstract:** From an aerodynamic point of view, the electric turbocharger for the air supply of an automotive fuel cell faces difficult requirements: it must not only control the pressure level of the fuel cell, but it also has to operate with very high efficiency over a wide range. This paper explores features for the compressor and the turbine of an existing electric turbocharger, which are intended to meet the specific requirements of a fuel cell in an experimentally validated numerical study. Adjustable diffuser or nozzle vanes in the compressor and turbine achieve wider operating ranges but compromise efficiency, especially because of the necessary gaps between vanes and end walls. For the turbine, there are additional efficiency losses since the pivoting of the nozzle vanes leads to incidence and thus to flow separation at the leading edge of the nozzle vanes and the rotor blades. An increase in the mass flow and a slight efficiency improvement of the turbine with the low solidity nozzle vanes counteracts these losses. For the compressor, a reduction in the diffuser height and its influence over the operating range and power consumption yields an increase in surge margin as well as in maximum efficiency.

**Keywords:** fuel cell air supply; operating range extension; electrical turbocharger

## 1. Introduction

The main components of fuel cell systems are the fuel cell itself, the cooling system, the hydrogen supply and the air supply. Figure 1 shows a typical design of the air supply system of an automotive fuel cell. Proton exchange membrane (PEM, also called polymer electrolyte membrane) fuel cells can be operated at either low or high pressures. Depending on the application, the air supply systems of PEM fuel cells use simple blowers with low pressure ratios or more complex centrifugal compressors with high pressure ratios. However, high system pressures are preferred in automotive applications because they increase the fuel cell system efficiency [1]. To estimate the power density of a PEM fuel cell, one uses, for example, the Nernst equation for the cathode of the fuel cell [2]:

$$E = E^0 - \frac{RT}{F} \cdot \ln\left(\frac{a_{OH^-}}{p_{O_2}/p^0}\right) \tag{1}$$

According to Equation (1), the electrode potential $E$ and thus the electrical output power of the fuel cell increases with the oxygen partial pressure $p_{O_2}$. The recommended operating pressures at the fuel cell design point are in a range between 2 and 4 bar [3–6]. However, Berning and Djilali [7] showed that the improvement in power density with oxygen partial pressure is less pronounced above operating pressures of 3 bar. Since the compressor of the cathode air supply system has the highest parasitic power consumption of the entire fuel cell system, a compromise must be found between the stack output power and compressor power consumption.

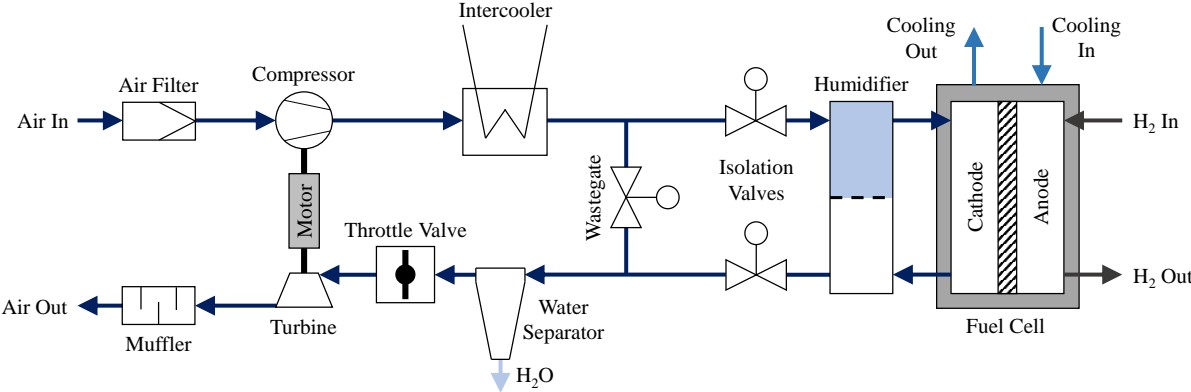

**Figure 1.** Typical design of the air supply system of an automotive fuel cell.

Since efficiency is of paramount importance, PEM fuel cell applications tend to use centrifugal compressors with vaned diffusers rather than vaneless diffusers. Nevertheless, while vaned diffusers achieve high efficiencies, they limit the operating range of the compressor [8]. Vaned diffusers are very sensitive to flow incidences and can be responsible for compressor surge. As explained previously, the fuel cell requires high pressure ratios at high mass flow rates but also often operates in the part-load range at very low mass flow rates [9]. Thus, the compressor surge-line should be shifted to low mass flow rates to prevent the compressor surge in the part-load range.

The diffuser stall is often initiated near the shroud in the vaneless space between the impeller and the diffuser [10]. The simplest way to influence the operating range of an existing centrifugal compressor with a vaned diffuser is to change the flow incidence or the flow velocity in the diffuser inflow. This is achieved by changing the leading edge angle or the vane height of the diffuser. Both measures influence the narrowest cross-section of the diffuser. Eynon and Whitfield [11] and Guandal and Gorvadhan [12] investigated the influence of the leading edge angle of the diffuser on the compressor map. They showed that a reduction in the leading edge angle results in a shift of the compressor operating range towards lower mass flow rates. By implementing an actuation mechanism, which allows the flexible pivoting of the diffuser vanes, the operating range of the compressor can be adjusted depending on the operating conditions [13,14]. However, gaps between end walls and diffuser vanes, which are necessary for a flexible pivoting concept, cause aerodynamic losses and thus reduce efficiency.

To further reduce the power consumption of the cathode air supply, a turbine was installed which uses the exhaust gas enthalpy of the fuel cell. According to Filsinger et al. [6], a turbine is able to recover more than 30% of the compressor power depending on the operating point. This turbine needs to fulfill the following important design requirements: a large operating range with high efficiency, so that power can be recovered over the entire operating range of the fuel cell, and a variable throttle device to adjust the fuel cell operating pressure. A variable nozzle turbine, which varies the narrowest flow cross-section by pivoting the vanes about their own axis meets these requirements [15]. Unfortunately, the required gaps between the end walls and the vanes will also be an issue.

The investigations of this paper are part of the research project ARIEL (German short form for: Charging of Fuel Cell Systems through Interdisciplinary Developed Electrically Driven Air Compressors) which aimes to optimize an existing electrically driven turbocharger (baseline design) for the cathode air supply of an automotive fuel cell [16]. Firstly, this paper numerically investigates the influence of the diffuser height as a compressor design parameter on the compressor map and efficiency. It is to be shown shown that the fuel cell-specific operating range can be achieved by a specific adjustment of this design parameter. The potential of the continuous pivoting of the compressor diffuser and the turbine nozzle vanes is to be numerically determined. Finally, the influence of the vane

solidity and the gap between the nozzle vanes and end walls of a variable nozzle turbine shall be analyzed.

## 2. Investigated Geometries and Parameters

### 2.1. Compressor

Previous investigations have shown that high positive flow incidences near the shroud in the diffuser inflow cause instabilities [17] which contribute to a compressor surge. These high positive incidences result from a strong wake region originating from the impeller tip-gap vortex. To reduce the influence of the impeller tip-gap vortex on the diffuser, the channel height $h_D$ between the impeller and diffuser (vaneless space) is continuously reduced (as can be seen in Figure 2). One of the objectives of this paper was to evaluate how the reduction in the diffuser height influences the compressor map and efficiency. The reduction in the diffuser height was combined with a redesign of the volute. Aungier [18] pointed out that the cross-sectional area of the volute at a certain circumferential position $\theta$ should be proportional to the diffuser outflow angle, as well as to the diffuser height at the outlet. We developed a new volute with a linear cross-sectional area progression and a smaller outlet area based on this finding. This volute was already used for the investigations of Schödel et al. [17]. A linear area progression of the volute achieves a uniform pressure field over the circumference, and thus a uniform diffuser load. Figure 3 shows the area progression of the baseline volute and the redesigned volute.

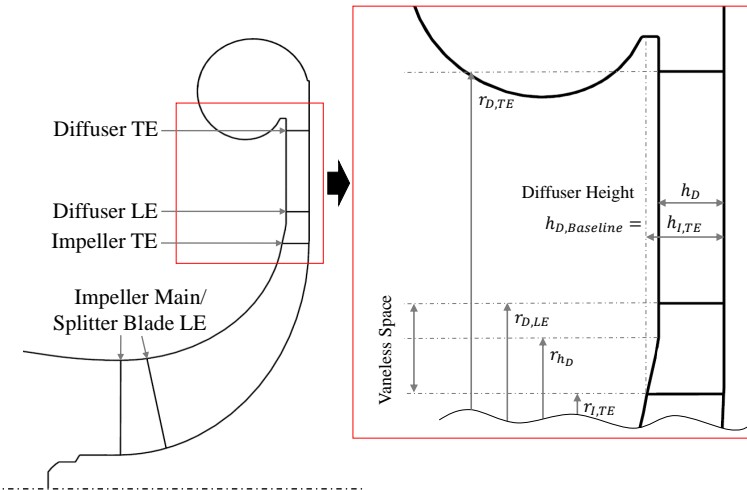

**Figure 2.** Reduction in diffuser height $h_D$.

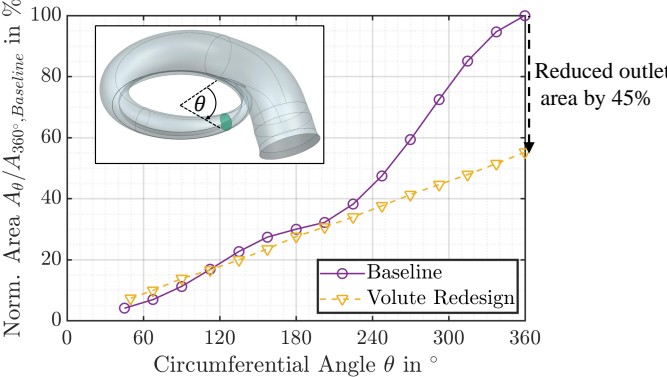

**Figure 3.** Comparison of the volute area development with the circumferential angle between the baseline volute and the redesigned volute (adapted from [17]).

The second analyzed configuration was a pivoting vane diffuser without gaps between the vanes and the end walls, as previously described and investigated by Schödel et al. [17] (as can be seen in Figure 4). The chord length of the baseline diffuser vanes was shortened for the pivoting vane diffuser. Pivoting the vanes around their central axis changes both the vane leading and trailing edge angle by the same amount. Negative pivot angles are expected for large parts of the compressor map, since negative pivot angles and thus reduced vane leading edge angles achieve a shift in the compressor surge line towards lower mass flow rates. Consequently, the trailing edge angle of the vanes also decreases for large parts of the compressor map. The redesigned volute (see Figure 3) was therefore also used for the pivoting vane diffuser. In the context of this work, the pivot angles between $-6°$ and $+2°$ were investigated.

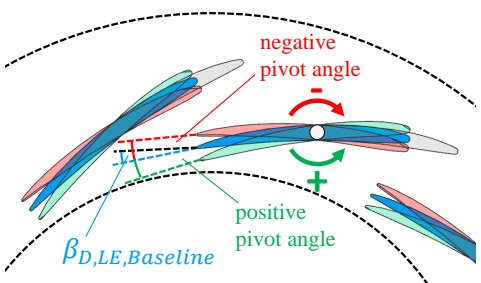

**Figure 4.** Pivoting vane diffuser (adapted from [17]).

*2.2. Turbine*

On the turbine side, the feature of a variable nozzle turbine (VNT) with only a tip gap considered at the shroud was investigated to evaluate the potential for extending the operating range. Figure 5 shows the rotation axis and the rotation directions of the pivoting mechanism. Menze et al. [19] already showed for this configuration that the VNT can result in a significant extension of the operating range for fuel cell application by positive pivot angles. In this study, the operating points for the VNT were derived from the extended turbine map.

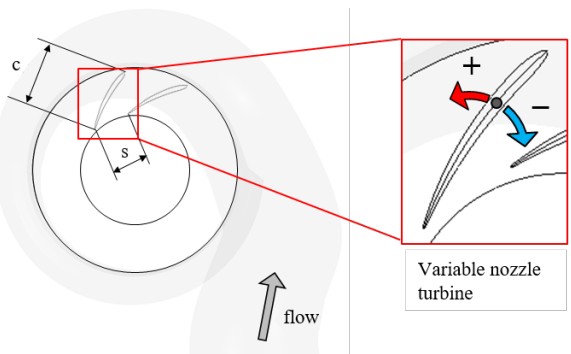

**Figure 5.** Definition of the rotation direction of a variable nozzle turbine and parameter visualization for the definition of solidity.

The derivation of operating points takes into account the speed coupling between the turbine and compressor which is given by the electrical drive of both components mentioned. Whenever using a VNT, gaps between the nozzle vanes and the end walls occur at both the hub and shroud. To evaluate the influence of these gaps, additional gap simulations of the baseline turbine at design speed are analyzed. For this purpose, different tip gap heights are specified in the area of the vaned nozzle. Since it was shown in Menze et al. [19] that the pivoting of the nozzle vanes is not sufficient to fully cover the operating range, an investigation of low solidity turbine geometries will be used to

evaluate the potential for extending the operating range but also for improving efficiency. The solidity $\sigma$ is defined as the ratio of chord length $c$ to pitch $s$ ( as can be seen in Figure 5):

$$\sigma = \frac{c}{s} . \tag{2}$$

In this study, the reduced solidity refers to the area of the vaned nozzle in which the number of vanes and the chord length are reduced in isolation in order to achieve the effect of reduced solidity. The blade angle and the radial position of the leading edges of the nozzle vanes remain unchanged compared to the baseline turbine.

## 3. Numerical Setup

### 3.1. Compressor

For the centrifugal compressor, steady-state simulations with the commercial software ANSYS CFX 19.2 were carried out. The numerical setup for the compressor investigations that is shown by Figure 6 was already described in more detail by Schödel et al. [17]. It consists of six domains: the inlet pipe, impeller, and backside cavity are modeled as single passage domains (only one angular segment) with periodic boundary conditions; whilst the diffuser, volute, and outlet pipe are modeled as full 360-degree domains to consider the circumferential asymmetry of the volute. The rotating domain impeller is connected to the inlet pipe and the diffuser by mixing plane interfaces (circumferential averaging of flow variables), whereas the connection between the impeller and backside cavity is implemented by a frozen rotor interface (direct mapping of flow variables). The total pressure and total temperature are chosen as inlet boundary conditions. The outlet boundary condition is specified by the mass flow rate. However, a second outlet with a mass flow boundary condition allowes one percent of the main mass flow rate to leave the backside cavity. Below, the described setup is referred to as the mixing plane setup. As a second setup, the frozen rotor setup was used for the experimental validation. In contrast to the mixing plane setup, the frozen rotor setup models all domains as 360° domains. Moreover, frozen rotor interfaces replace the mixing planes between the inlet pipe and impeller as well as between the impeller and diffuser [17].

Since the frequently used Menter $k$-$\omega$ SST turbulence model [20] often overestimates flow separations, this paper employs the Wilcox $k$-$\omega$ turbulence model [21]. Convergence to RMS residuals smaller than $10^{-4}$ and imbalances smaller than 0.05% ensure the stability of steady-state simulations. In addition, the negative slope of the speed lines in the compressor map and the negative slope of the pressure rise coefficient between the impeller outlet and diffuser throat are used to prove the stability of an operating point [17].

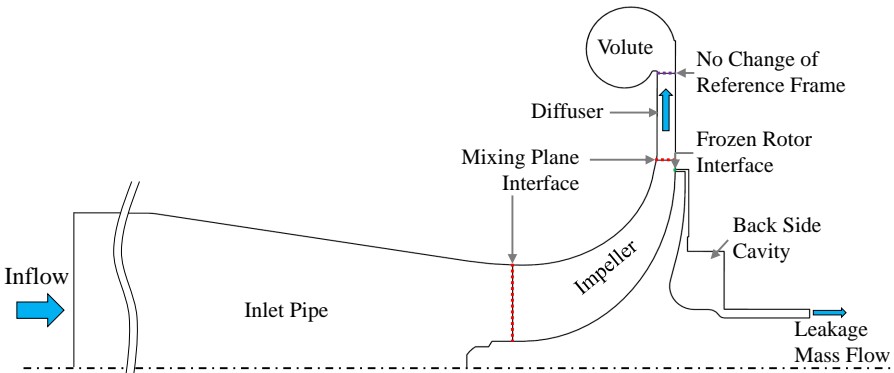

**Figure 6.** Numerical setup of the compressor simulations (mixing plane setup) (adapted from [17]).

### 3.2. Turbine

As for the centrifugal compressor, steady-state simulations were performed for the radial turbine using the software ANSYS CFX 19.2. The numerical setup for the investigations of the turbine shown in Figure 7, was already described in more detail by Menze et al. [19].

The numerical setup consists of four domains: volute, vaned nozzle, turbine rotor, and a diffuser outlet pipe. In order to better account for the asymmetry of the volute and its flow conditions, the volute and the vaned nozzle are modeled as full 360-degree domains. In contrast, the turbine rotor and the diffuser outlet pipe are modeled as single passages with periodic boundary conditions. The tip gap at the shroud of the vaned nozzle has a height of 2.1% of the nozzle passage height. For the interfaces between the vaned nozzle and the rotating domain of the turbine rotor as well as between the turbine rotor and the outlet pipe, a frozen rotor approach is applied. The total pressure and static temperature are specified as boundary conditions at the inlet, while static pressure is set at the outlet. The static temperature at the inlet is set to 80 °C to represent typical inlet conditions for turbines in fuel cell applications. The Menter *k-ω SST* model [20] is employed for turbulence modeling.

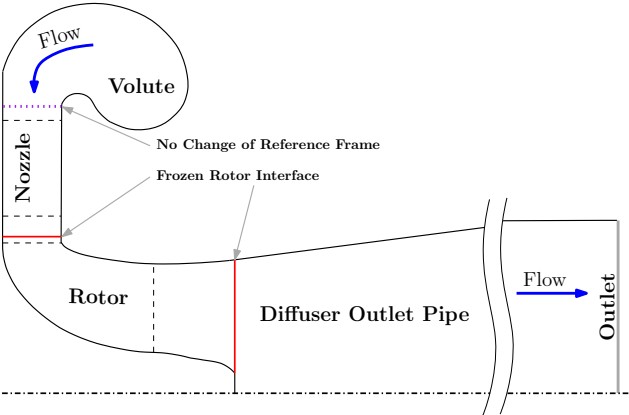

**Figure 7.** Numerical setup of the turbine simulations (adapted from [19]).

## 4. Results

*4.1. Compressor*

4.1.1. Experimental Validation

Figure 8 compares the experimental results and the CFD (short form for: Computational Fluid Dynamics) results which were generated with the mixing plane and the frozen rotor setup for the design speed and 60% of the design speed. The quantities in Figure 8 and the compressor maps below are specified relative to the CFDs' best efficiency operating point at design speed (mixing plane setup) [17].

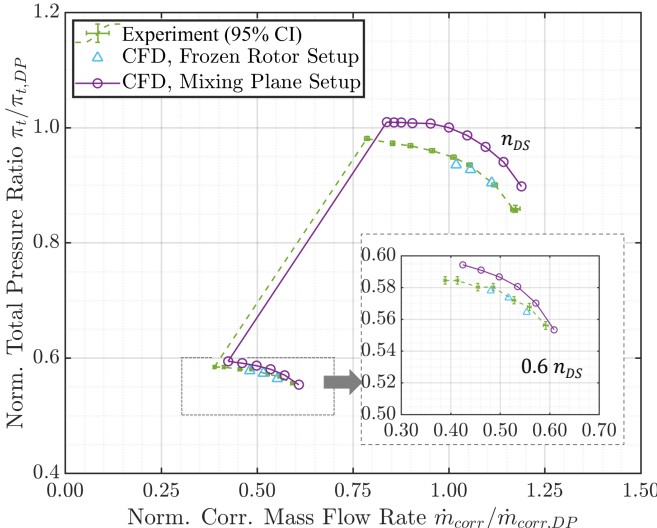

**Figure 8.** Comparison between the experimental (provided by the Volkswagen AG) and CFD results (mixing plane and frozen rotor setup) for the baseline compressor (adapted from [17]).

The CFD results obtained with the frozen rotor setup agree very well with the experimental results. However, one observes a significant overestimation of the total pressure when the mixing plane setup is compared to the experimental results, especially at design speed. These deviations are caused by the mixing plane itself, which reduces interactions between the diffuser and impeller. In addition, the non-uniform volute pressure field has less influence on the impeller flow due to the mixing plane. Regarding the experiment or the frozen rotor setup, the interactions between the blade rows and non-uniformity lead to further total pressure losses in the impeller, diffuser, and volute [17].

Although the total pressure ratios between the experimental and the numerical results of the mixing plane setup differ significantly in some operating points, the deviations between the surge lines are rather small. Since numerical instabilities near surge prevent the convergence of steady-state CFD simulations, the experimental surge line usually occurs at lower mass flows. Transient CFD simulations with very high computational effort would be necessary to identify the surge line exactly. It is assumed that the shift between the experimental and numerical surge line remains constant when the diffuser geometry is changed. Future experimental studies with different diffuser geometries are planned to verify this assumption [17].

### 4.1.2. Baseline Compressor

The map of the compressor has been partly shown by Figure 8. Figure 9 visualizes the entire map together with other configurations and the fuel cell operating line. The fuel cell operating line does not seem to be completely covered. A full coverage of the fuel cell operating line is only achieved for normalized corrected mass flow rates greater than 0.70 with an assumed surge margin of 15%. The surge margin was calculated by the corrected mass flow rate at the fuel cell operating point (OP) and the corrected mass flow rate near stall (NS) according to Equation (3) at a constant total pressure ratio:

$$SM_{\Pi_t=const} = \frac{\dot{m}_{corr,OP}}{\dot{m}_{corr,NS}} - 1 \qquad (3)$$

This surge margin allows a simple evaluation along the entire fuel cell operating line. For mass flow rates less than 0.70, a bypass control is needed in order to meet the fuel cell requirements while safely operating the compressor. This means that part of the compressed air is extracted through a control valve, also called wastegate (as can be seen in Figure 1), and directed to the turbine, thus bypassing the fuel cell. As a consequence, the compressor at low speeds is operated at higher mass flow rates than required, thus consuming more power since power is proportional to the mass flow rate.

### 4.1.3. Reduction in the Diffuser Height

The volute redesign was considered first in order to be able to evaluate the influence of the volute redesign and the change in diffuser height separately. As already explained by Schödel et al. [17], the volute redesign only has a minor influence on the compressor operating range. Due to the new volute, however, the polytropic efficiency increases over the entire operating range. The maximum efficiency at the different speed lines increases by 0.80 to 0.98 percentage points. The reduced volute size leads to a reduction in flow separations that occur in the diffuser and the volute. Furthermore, the linear area progression achieves a more uniform pressure field around the circumference.

As a next step, Figure 9 shows how reducing the diffuser height and thus the narrowest diffuser cross-section influences the compressor map. The smaller the diffuser height, the more the surge and choke line shift towards lower mass flow rates. Compared to the baseline compressor, the surge margin increases by up to 23% at 60% of the design speed according to Equation (3). If a required surge margin of at least 15% is considered, a full coverage of the fuel cell operating line (without the bypass control) for normalized corrected mass flows greater than 0.32 will be achieved by a 10% reduction in the diffuser height. Thus, the compressor operating line shifts towards lower mass flows compared to

the baseline compressor between mass flow rates of 0.32 and 0.70. This reduces the power consumption of the compressor in this range.

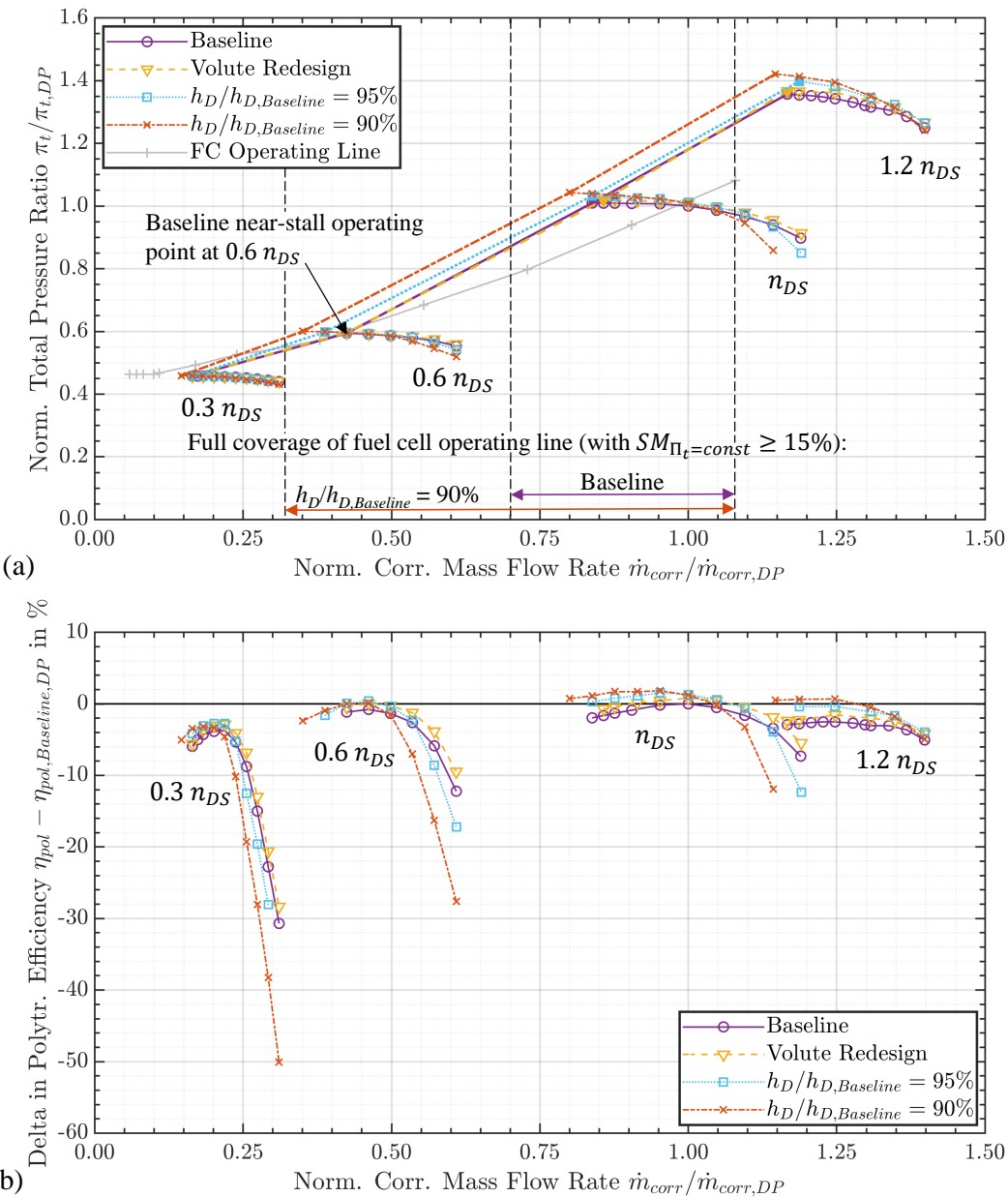

**Figure 9.** Compressor map (**a**) and polytropic efficiency (**b**) for the baseline compressor, a volute redesign, and a reduction in the diffuser height.

The efficiency plot in Figure 9 shows that the polytropic efficiency near the surge line increases when the diffuser height decreases, especially at higher speeds. The potential of a 10% reduction in the diffuser height is estimated from the change in efficiency and the shift of the compressor operating line towards smaller mass flow rates at lower speeds: at 60% of the design speed, one can observe the potential to reduce the required power of the compressor by up to 18%. However, even at design speed, there is the potential to reduce the required compressor power by up to 2%.

To further evaluate how a reduction in the channel height influences the diffuser flow, the radial mass flow density upstream of the diffuser vanes and the diffuser incidence in Figure 10a,b are considered.

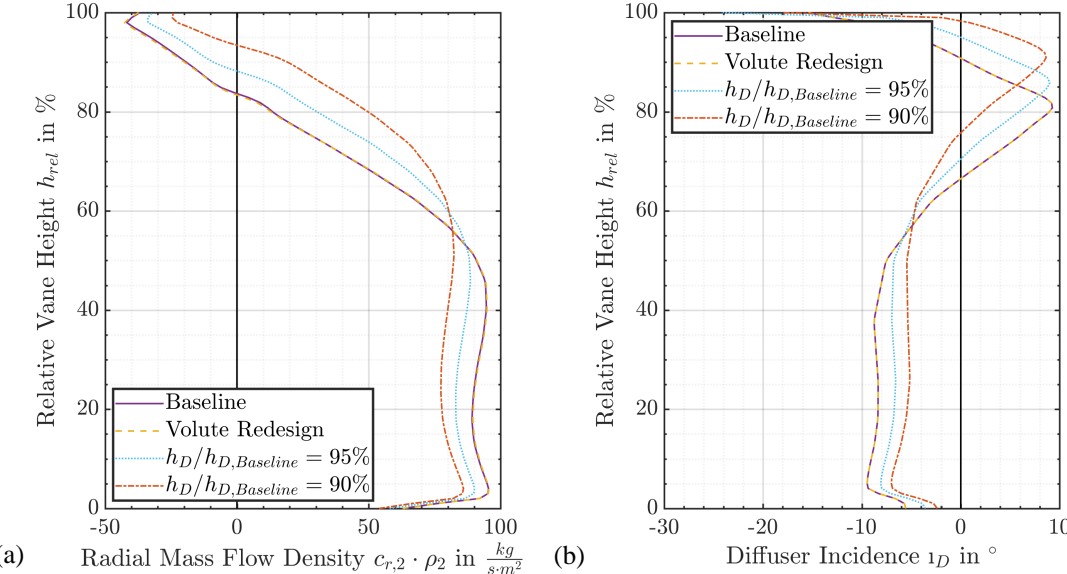

**Figure 10.** Radial mass flow density (**a**) and incidence (**b**) in the diffuser inflow for the baseline compressor, a volute redesign, and a reduction in the diffuser height at the baseline near the stall operating point at 60% of the design speed.

For a 10% reduction in the diffuser height, the extent of the region with negative radial mass flow density, which indicates a backflow near the diffuser shroud, decreases from 16% to 6% of the diffuser vane height. The backflow region results from the impeller tip gap vortex and decreases as the diffuser height is reduced because the flow near the diffuser shroud accelerates compared to the baseline diffuser. Consequently, the diffuser's incidence and thus the tendency towards instability decreases in the region between 57% and 85% of the diffuser vane height. A decrease in the radial mass flow density below 56% of the diffuser vane height results due to the conservation of mass flow. The diffuser incidence, which is negative in this region, decreases in amount. As shown above, this leads to an increase in the polytropic compressor efficiency.

### 4.1.4. Pivoting Vane Diffuser

As a comparison, the potential of a pivoting vane diffuser is analyzed. Figure 11 shows the performance map and the polytropic efficiency of the compressor with pivoting diffuser vanes, with no gaps between end walls and diffuser vanes. The pivot angle was chosen for the different operating points such that the maximum operating range and efficiency is achieved.

It is possible to achieve a large operating range with the pivoting vane diffuser since the diffuser leading edge angle and thus the incidence can be continuously adjusted. The surge margin increases by up to 53% at 60% of the design speed and the fuel cell operating line is fully covered with a surge margin of at least 15% for normalized corrected mass flow rates above 0.20. If the diffuser gaps are neglected, the efficiency is maintained or even improved.

Nevertheless, Schödel et al. [17] showed that gaps of 2.8% of the vane height at both the hub and the shroud, led to a decrease in compressor efficiency over the entire operating range. For a pivot angle of 0°, the peak efficiency decreases by 2.23 percentage points. If we now consider the changed operating line and efficiency for the pivoting vane diffuser (including gaps) compared to the baseline compressor, it is possible to estimate the change in performance for this configuration: while at 60% of the design speed, the potential to decrease the required power for driving the compressor by up to 12% exists, at 100% of the design speed, the power consumption increases by approximately 3%.

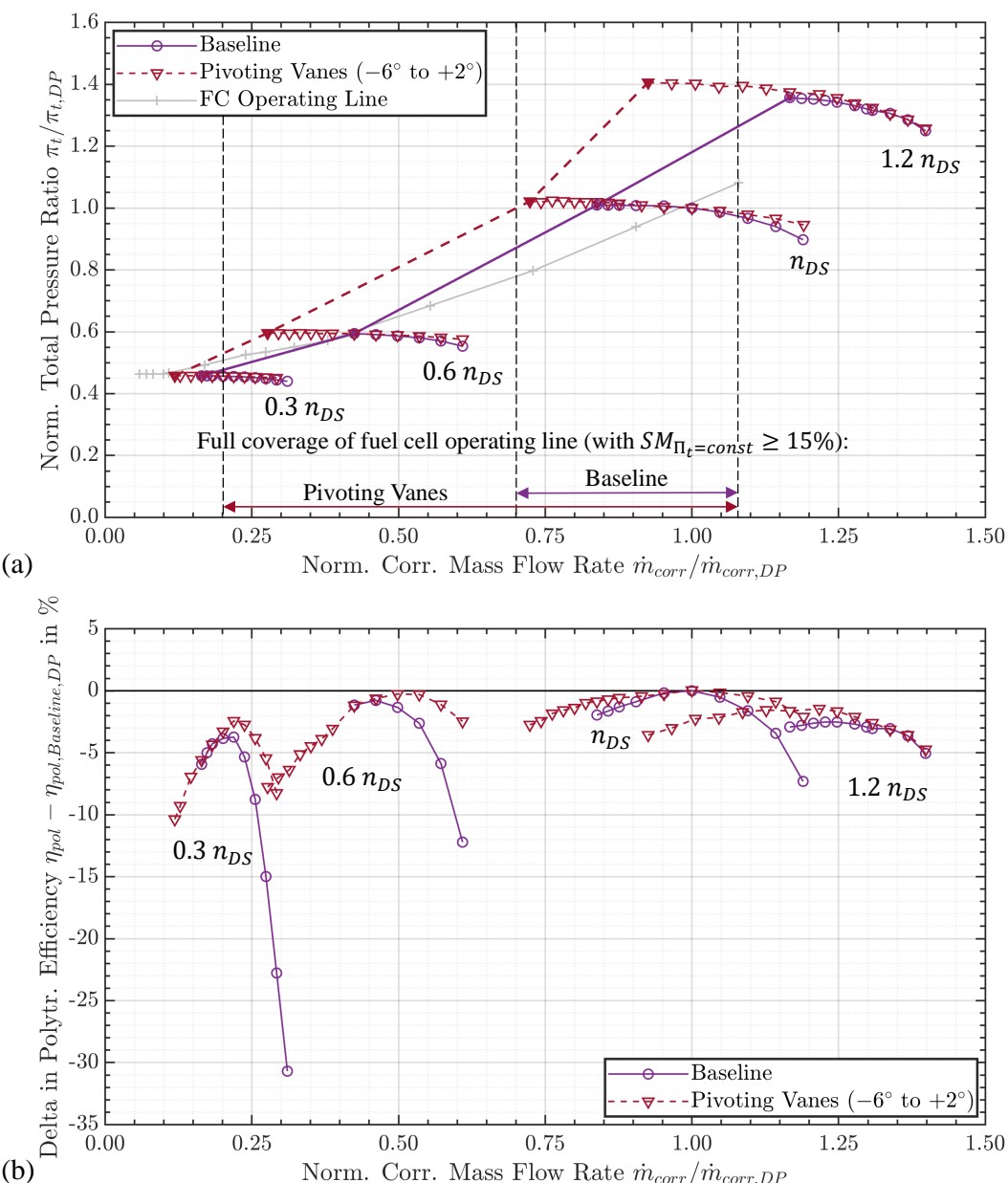

**Figure 11.** Compressor map (**a**) and polytropic efficiency (**b**) for the baseline and a pivoting vane compressor.

### 4.2. Turbine

#### 4.2.1. Experimental Validation

Figure 12 shows the experimental and CFD results of the baseline turbine for two different speeds in terms of the normalized corrected mass flow and the normalized isentropic total efficiency vs. the normalized total pressure ratio. The quantities shown in this and the following figures are normalized with the values of the turbine simulated with the mass flow as a boundary condition taken from the best efficiency operating point of the compressor at design speed.

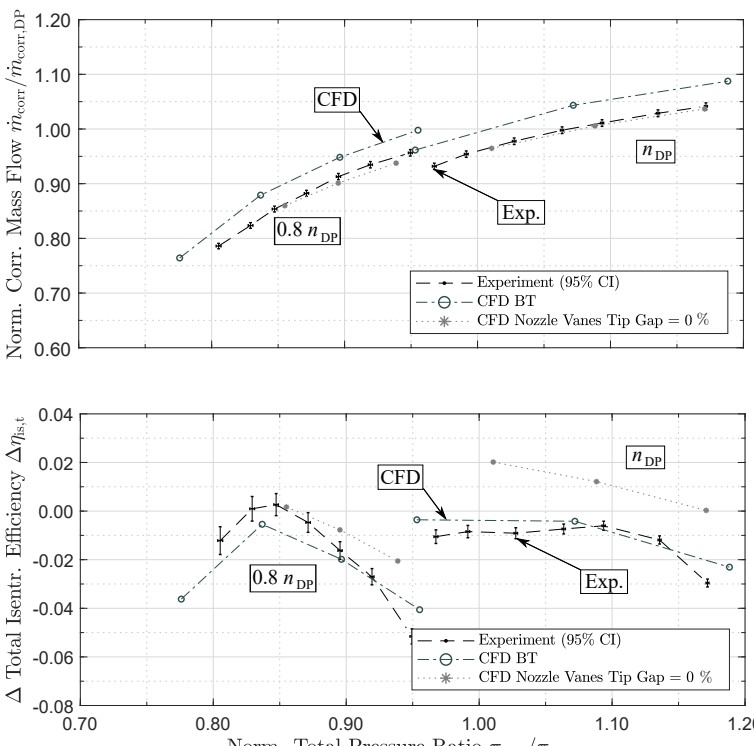

**Figure 12.** Comparison between the CFD and experimental (provided by Volkswagen AG) results of the baseline turbine (adapted from [19]).

The experimental and CFD results show an offset of 3% in the normalized mass flow relative to the real mass flow in the experiment. These discrepancies are explained by a different nozzle passage height between the experimentally investigated turbine and the simulation model. In the simulation, a tip gap height corresponding to the exact manufacturing specifications of 2.1% is modeled. If this tip gap and the vane height of the nozzle passage are smaller due to manufacturing tolerances, the flow cross-section is also reduced, which explains the aforementioned offset. This was confirmed with additional simulations. These results are also shown in Figure 12. Further minor influences as well as information on the experimental validation were described in Menze et al. [19].

### 4.2.2. Variable Nozzle Turbine

Menze et al. [19] already showed that a significant extension of the operating range can be achieved using the VNT. However, the complete operating range cannot be covered, especially not in the range of low pressure ratios. In Menze et al. [19], the turbine was considered in isolation, neglecting the coupling of the turbine and compressor via an electric motor, which resulted in the same speed for both components at each operating point. This coupling will be taken into account in the present CFD study as shown below.

Figure 13 shows the turbine map and the normalized isentropic efficiency of the variable nozzle turbine for three adjustment positions of the VNT. The operating line of the fuel cell is also shown. Furthermore, four operating points and the operating range with speed coupling are highlighted. It is shown that operating points for the fuel cell can be adjusted for a normalized total pressure ratio above 0.74, which is a reduction of approximately 7% compared to Menze et al. [19] because of the speed coupling. For the marked operating points, boundary conditions for the operation of a fuel cell are interpolated and derived based on the simulated data. Table 1 shows the necessary speeds and VNT areas for adjusting the correct mass flow and pressure ratio for the operation. In addition, an estimate of the efficiency is given. This shows that the turbine has significantly reduced efficiencies at lower pressure ratios. This can be explained by the strong pressure-

sided flow of the guide vanes due to the positive pivot angle of the vanes and the resulting flow separation.

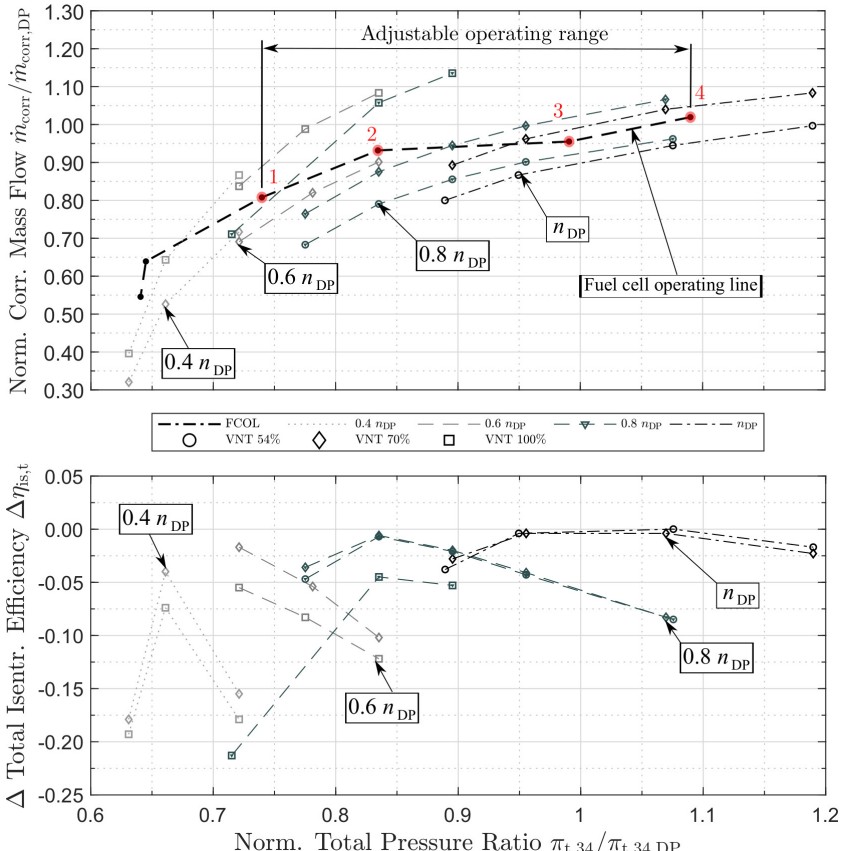

**Figure 13.** Turbine map with the derived operating points and total isentropic efficiency for the VNT.

**Table 1.** Derived operating points of the VNT.

| Operating Point | $n_{corr}$ | $\dfrac{\pi_{t,34}}{\pi_{t,34,DP}}$ | $\dfrac{\dot{m}_{corr}}{\dot{m}_{corr,DP}}$ | VNT Area in % | $\Delta\eta_{is,t}$ |
|:---:|:---:|:---:|:---:|:---:|:---:|
| 1 | $0.72 \cdot n_{DP}$ | 0.740 | 0.808 | ~98 | −0.12 |
| 2 | $0.88 \cdot n_{DP}$ | 0.835 | 0.932 | ~85 | −0.03 |
| 3 | $n_{DP}$ | 0.991 | 0.955 | ~65 | −0.01 |
| 4 | $1.08 \cdot n_{DP}$ | 1.090 | 1.019 | ~65 | 0 |

For pressure ratios lower than 0.74, a wastegate valve is needed so that the larger mass flow required by the fuel cell can flow through, and the correct operating point can be adjusted. Since a wastegate results in losses due to the turbine being by-passed, the objective is to have a variable turbine which can cover the full operating range. One way to achieve this could be to reduce the solidity of the turbine. After first examining the influence of tip gaps at the hub and shroud, the potential of the aforementioned measure is then analyzed in more detail.

### 4.2.3. Influences of Tip Gaps

Figure 14 shows the normalized corrected mass flow and the normalized isentropic efficiency over the normalized total pressure ratio for the baseline turbine with tip gap variations at design speed. In addition to the baseline turbine with a tip gap height of 2.1% at the shroud, the numerical results of two variations with 1.4% and 2.8% tip gap height (TG) at the shroud and hub are shown. Figure 14 indicates that the mass flow increases

with the tip gap height compared to the baseline turbine, since the tip gap has the effect of widening the flow cross-section. This allows a larger mass flow to pass through the turbine at the same pressure level. For a tip gap height of 2.8%, there is a maximum mass flow increase of approximately 4.3% at a normalized total pressure ratio of 0.95. The efficiency distribution basically shows that the efficiency decreases with increasing gap height. A maximum efficiency deficit of approximately 4.3 percentage points occurs at a normalized total pressure ratio of 1.19 for a tip gap height of 2.8%. Additionally, it is noticeable that as the total pressure ratio increases, the efficiency loss becomes larger. This can be attributed to the increasing tip gap mass flow from the pressure side to the suction side. Consequently, when using a VNT for operating point adjustment, significant efficiency losses due to gap losses are to be expected. These losses should be taken into account in future simulations. However, a smaller tip gap than 2.8% is certainly feasible by design. As a consequence, the resulting efficiency losses are lower than the maximum efficiency loss shown previously.

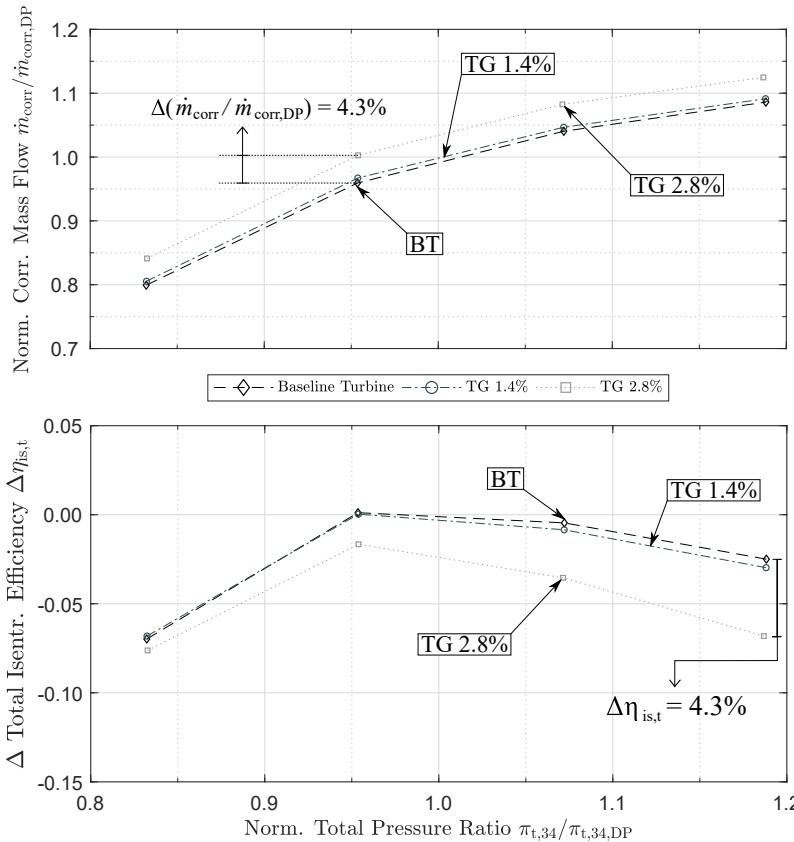

**Figure 14.** Influence of tip gap variations on the flow factor and isentropic efficiency at design speed.

### 4.2.4. Low Solidity Turbine

In the following section, the influence of a reduced solidity turbine is investigated. This is achieved by either reducing the number of nozzle vanes $z$ or the chord length $c$ in the area of the vaned nozzle (see Figure 5). These modifications were applied to the baseline turbine. Figure 15 shows the normalized corrected mass flow and the normalized total isentropic efficiency for two variations of each of the aforementioned variables at two different speeds which were compared with the baseline turbine. The purpose of the variations is to increase the flow cross-section; therefore, the number of vanes is reduced and the chord length of the vaned nozzle is shortened.

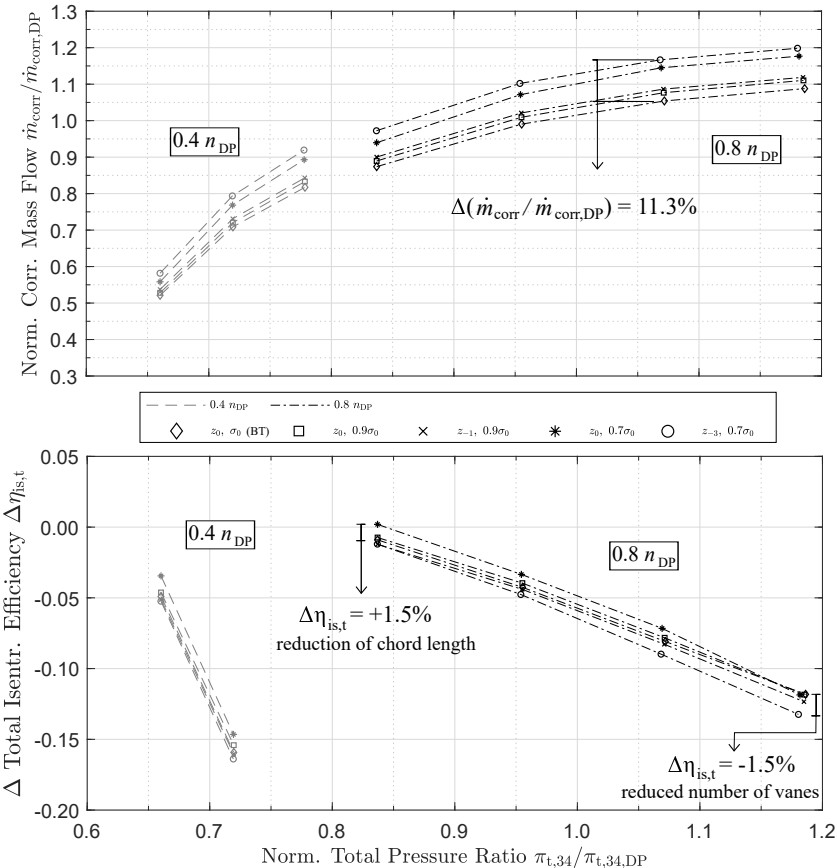

**Figure 15.** Turbine map and isentropic efficiency for the baseline and the low solidity turbine.

For the cases investigated, there is basically an increase in the mass flow with a decrease in solidity at the same pressure ratio. The reason for this is the increase in the narrowest flow cross-section due to the geometric changes. As expected, the mass flow increases approximately proportionally to the flow cross-section according to the continuity equation. When the number of vanes is reduced, the mass flow increases by up to 11.3% in the cases investigated, and when the chord length is reduced, an increase of up to 9.1% can be seen each at a normalized total pressure ratio of 1.07. This discrepancy can already be explained by the fact that the flow cross-section is larger for vane reduction than for reduced chord length—despite them having the same solidity.

Compared to the baseline turbine, a reduced number of vanes slightly decreases the efficiency with losses of up to 1.5 percentage points. This effect can be explained by two phenomena. Firstly, there is a decreased flow deflection due to a lower guidance of the flow through the vanes. In addition, the increased mass flow rate leads to an increase in the absolute inflow velocity at the rotor inlet. Both effects cause a change in the inflow of the impeller to a pressure-sided inflow at unchanged speed, which lead to separation losses. Furthermore, the geometric variation results in a larger total pressure loss in the vaned nozzle. For the reduction in the chord length, however, there is a slightly improved efficiency of up to 1.5 percentage points compared to the baseline turbine. Due to the shortened nozzle vanes, the flow is deflected in the radial direction after leaving the vaned nozzle, whereby this effect positively counteracts the increased absolute inflow velocity at the inlet of the rotor. In addition, the simulations with reduced chord length show a lower total pressure loss across the vaned nozzle.

Due to the potential for increasing the mass flow rate by the presented geometrical modifications, a positive effect in terms of operating range coverage can also be estimated when applied to a VNT. Therefore, the demonstrated influence of the reduced solidity with modified chord length will be further investigated in future experimental studies.

## 5. Conclusions

To meet the specific requirements of the fuel cell, it is necessary to develop electrical turbochargers for the cathode air supply system which offer a large operating range together with high efficiency. Reducing the diffuser height is a simple measure which shifts the compressor operating range towards lower mass flow rates and achieves a better coverage of the operating line of an automotive fuel cell. Continuously decreasing the diffuser height on the shroud side in the vaneless space reduces the backflow in the diffuser inflow which results from the rotor tip-gap vortex. Consequently, the flow at the leading edge of the diffuser vanes becomes more uniform over the diffuser height and the amount of incidence decreases. Overall, this increases the efficiency of the compressor. The reduction in the diffuser height has the potential to improve the compressor performance for its use in fuel cell applications at both high and low speeds.

A pivoting vane diffuser, which requires a mechanically complex actuation mechanism, covers the operating line of the fuel cell even better. The gaps between the diffuser vanes and the end walls nevertheless lead to a large loss of efficiency. Although pivoting vanes improve the performance of the compressor at low speed, they reduce the performance at high speed.

In addition to the requirements above, the turbine ideally uses a variable throttle to flexibly control the operating pressure of the fuel cell. The variable nozzle turbine shows a significantly improved coverage of the fuel cell operating line. The derived operating points show the potential of an operation along the operating line from a normalized total pressure ratio of 0.77. However, adjusting the operating range by pivoting the nozzle vanes causes significant efficiency losses, especially due to the incidence in the rotor inflow. Furthermore, gaps at the end walls of the vanes lead to increased tip gap leakage losses and thus reduce efficiency. A reduction in the solidity by reducing the vane chord length shows the potential for providing larger mass flows due to the increased cross-sectional area with slightly improved efficiency. This provides the potential for a further improvement in operating range, or a reduction in the loss of efficiency due to the pivoting of the nozzle vanes. In summary, a VNT is suitable for controlling the operating pressure and mass flow for a fuel cell application.

## 6. Outlook on Experimental Investigations

This section shall provide an outlook on further experimental investigations of the electrical turbocharger within the ARIEL research project. The experimental investigations will be used to validate the potential of the pivoting vane diffuser and the VNT, among others. In addition to static pressure and temperature probes, transient static pressure sensors are also installed to determine the beginning of compressor stall. Furthermore, a transient evaluation of the turbocharger power consumption will be used to derive conclusions about the onset of instabilities.

Figure 16 visualizes the experimental setup. To set a constant inlet pressure and temperature, an air conditioning unit, which includes a volumetric flow measurement, is implemented at the suction end of the air supply of the compressor. A throttle valve (compressor back pressure unit) is necessary at the compressor outlet in order to adjust the operating point of the compressor or turbine. Furthermore, a water-cooled heat exchanger is installed upstream of the turbine to control the turbine inlet temperature. Finally, the turbine exhaust air leaves the test bench in the environment. The compressor inlet and outlet as well as the turbine inlet and outlet were instrumented for static pressure and temperature measurements. The test bench automation controls the settings of the DC voltage source, the power electronics, the water conditioning unit, the suction air conditioning unit, and the compressor back pressure unit.

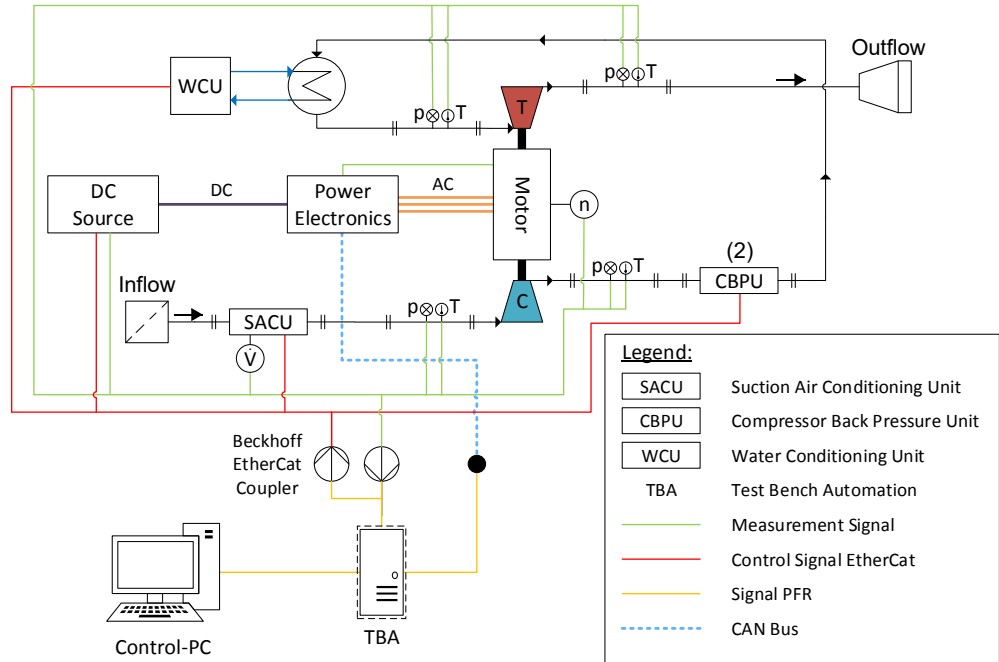

**Figure 16.** Schematic setup for future experimental investigations.

**Author Contributions:** Conceptualization, M.S., M.M. and J.R.S.; methodology, M.S. and M.M.; validation, M.S. and M.M., investigation, M.S. and M.M.; writing—original draft preparation, M.S. and M.M.; writing—review and editing, M.S., M.M. and J.R.S.; visualization, M.S. and M.M.; supervision, J.R.S.; project administration, M.S. and M.M.; funding acquisition, J.R.S. All authors have read and agreed to the published version of the manuscript.

**Funding:** This research was funded by the Federal Ministry of Transport and Digital Infrastructure (BMVI) grant number 03B10105C/2.

**Institutional Review Board Statement:** Not applicable.

**Informed Consent Statement:** Not applicable.

**Data Availability Statement:** Data sharing not applicable.

**Acknowledgments:** The investigations presented in this paper are part of the research project "Charging of Fuel Cell Systems through Interdisciplinary Developed Electrically Driven Air Compressors" (ARIEL). The authors thank the Federal Ministry of Transport and Digital Infrastructure (BMVI) which financially supported the work within the framework of the National Innovation Programme (NIP) Hydrogen and Fuel Cell Technology as well as the NOW GmbH which coordinated the funding guideline. The authors would also like to gratefully acknowledge the entire project consortium consisting of the Volkswagen AG and all the participating institutes of the University of Braunschweig and the Ostfalia University of Applied Sciences.

**Conflicts of Interest:** The authors declare no conflict of interest.

## Abbreviations

The following symbols and abbreviations are used in this manuscript:

| | | | |
|---|---|---|---|
| a | Activity | A | Area |
| c | Chord Length | $c_{r,2}$ | Radial Flow Velocity At Diffuser Inlet |
| E | Electrode Potential | $E^0$ | Normal Potential |
| F | Faraday Constant | h | Height |
| $h_{rel}$ | Relative Height | i | Incidence |
| $\dot{m}_{corr}$ | Corrected Mass Flow Rate | $p_{O_2}$ | Oxygen Partial Pressure |

| $p^0$ | Standard Pressure | r | Radius |
|---|---|---|---|
| R | Universal Gas Constant | s | Pitch |
| SM | Surge Margin | $\eta_{\text{is}}$ | Isentropic Efficiency |
| $\eta_{\text{pol}}$ | Polytropic Efficiency | $\theta$ | Circumferential Angle |
| $\Pi_t$ | Total Pressure Ratio | $\rho_2$ | Density at Diffuser Inlet |
| $\sigma$ | Solidity | ARIEL | Charging of Fuel Cell Systems through |
| CFD | Computational Fluid | | Interdisciplinary Developed Electrically |
| | Dynamics | | Driven Air Compressors |
| CI | Confidence Interval | const | Constant |
| D | Diffuser | DC | Direct Current |
| DP | Design Point | DS | Design Speed |
| FC | Fuel Cell | I | Impeller |
| LE | Leading Edge | MDPI | Multidisciplinary Digital Publishing Institute |
| NS | Near Stall | OP | Operating Point |
| PEM | Polymer Electrolyte Membrane | PEMFC | Polymer Electrolyte Membrane Fuel Cell |
| TE | Trailing Edge | VNT | Variable Nozzle Turbine |

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
