# Peer review of "Experimentally Validated Extension of the Operating Range of an Electrically Driven Turbocharger for Fuel Cell Applications"

_machines, doi:10.3390/machines9120331_

Round 1
Reviewer 1 Report
The authors are to be congratulated on their interesting and very well presented paper. My only suggestion refers to the abstract where you have written, ... , there are additional efficiency losses since pivoting of the nozzle vanes leads to incidence ... Should incidence be qualified, as say, excessive incidence?
Author Response
Thank you for your feedback and your comment. However, it will not be implemented at this point, as it is only intended to point out in general terms what primarily causes the losses. Depending on the operating point, the influence is smaller or greater.
Reviewer 2 Report
COMMENTS:
This paper focuses on using experiment to validate some features for the compressor and the turbine of an existing electric turbocharger. The manuscript discussed how height and pivoting vane of the diffuser in the compressor influence the compressor map and efficiency. How the variable nozzle vanes and the solidity in the turbine achieve wider operating ranges but also compromise the efficiency.
The writing and the research purpose are clear, and the topic fits the scope of the journal. The main work focuses on experimental validation. However, this manuscript is not yet ready for publication due to some mistakes and inadequate explanation about some arguments:
- In Line 34, on Page 1, it is mentioned that “such diffusers achieve high efficiencies, they limit the operating range of the compressor”, what does “such diffuser” mean? It stands for the vane diffusers or vaneless diffusers, it should be clear. If it means vane diffuser, the feature described here is not same as the following content of this manuscript.
- The background introduction is inadequate, the connection relationship between turbocharger and fuel cell or other components is not clear. The turbocharger’s structure is also not clear too. A systematic structural diagram or actual system picture is needed for better understanding.
- In Figure 1, which symbol refers to the diffuser height? And what does exactly the diffuser height mean?
- In Line 95, on Page 3, what does “pivot angles” mean? Is there any figures to better show its structure.
- In Line 128, on page 4, what is the difference between “mixing plane interfaces” and “frozen rotor interface”? More explanation on it should be given?
- In Line 151, on page 5, what is the difference between “full 360-degree domains” and “as single passages”? It should be clarified?
- As for equation 3, what does the subscript “OP” and “NS” below the mass flow rate exactly represent for?
- In Figure 7(b), the meaning of the ordinate is unclear, which orientation stand for more efficient, the positive direction or to the negative direction, it is confusing?
- In Line 223, on page 8, as for the observed conclusion from Figure 7, which data point of mass flow rate and diffuser height can exactly draw this specific conclusions?
- In Line 231, on page 8, it is mentioned that “the diffuser’s incidence and thus the tendency to instability decreases in the region between 57% and 85% of the diffuser vane height”, why does the diffuser’s incidence and the tendency to instability in this region decrease? Is that reasonable?
- Because the influence of the interaction between compressor and turbine on the operating line and efficiency of the fuel cell is not discussed systematically in this manuscript, it is unclear that how could we control the compressor and turbine to full cover fuel cell operating line? More discussion about that should be added.
Author Response
Thank you for your feedback. Please find comments and answers attached. I have also attached a revised version of the paper in which changes are marked.

Reviewer 3 Report
Dear authors,
I appreciate your efforts involved in this research paper. The topic is fascinating for the readers and the specialists working in the automotive fuel cell field. Interestingly, the investigations on the geometries and the parameters of the compressor and the turbine are part of an exciting research project ARIEL. This project aims to optimize an existing electrically-driven turbocharger (baseline design) for the cathode air supply of an automotive fuel cell. Based on a numerical setup selection for the centrifugal compressor and the radial turbine, several steady-state simulations are performed using the commercial software package ANSYS CFX 19.2. The experimental validation is based on the simulation results clearly and well presented in figures 6 to 13. These simulation results support the conclusions—also, an outlook on further experimental investigations of the electrical turbocharger within the ARIEL research project. To further increase this research paper's overall quality and visibility, I suggest you improve the Introduction and Reference sections by adding at least 3-4 updated documents to the most recent field literature (i.e. 2020-2021).
Author Response
Thank you for your feedback. Some updated documents (2018-2021) have been added to the introduction and reference section (see attached PDF with marked changes).
